# Symptomatic Uterine Rudiments in Adolescents and Adults with Mayer–Rokitansky–Küster–Hauser Syndrome (MRKHS): Management and Outcomes

**DOI:** 10.3390/jcm13226767

**Published:** 2024-11-10

**Authors:** Maria Buda, Weronika Zajączkowska, Klaudyna Madziar, Witold Kędzia, Karina Kapczuk

**Affiliations:** 1Gynaecological and Obstetric Clinical Hospital of Poznan University of Medical Sciences, Polna 33, 60-535 Poznan, Poland; 2Physiology Department, Poznan University of Medical Sciences, 60-781 Poznan, Poland; 3Division of Gynaecology, Poznan University of Medical Sciences, Polna 33, 60-535 Poznan, Poland

**Keywords:** Mayer–Rokitansky–Küster–Hauser syndrome, MRKH syndrome, Müllerian anomalies, rudimentary horn, uterine remnant, endometriosis

## Abstract

**Background:** Women with an aplastic uterus (ESHRE/ESGE classification) or Müllerian agenesis (ASRM MAC 2021) might present with functional uterine remnants. Our study aimed to report the clinical course of symptomatic uterine rudiments in adolescents and adults with Mayer–Rokitansky–Küster–Hauser syndrome (MRKHS). **Methods:** This study involved 20 patients with MRKHS who, between 2012 and 2023, underwent surgery for symptomatic uterine horns at the mean age of 25.2 years in the Division of Gynaecology, Gynaecological and Obstetric Clinical Hospital, Poznan University of Medical Sciences, Poland. The records of the patients were retrospectively analysed. **Results:** The volume of the uterine horns ranged from 0.8 to 58.3 cm^3^, and the volume of the endometrial cavity within the horns ranged from 0.03 to 12 cm^3^, with no significant difference between adolescents and adults (*p* = 0.36). In five patients (25%), MRKHS was identified during the diagnosis of recurrent pelvic pain at the age of 12.6–14.8 years. In 19 patients, uterine rudiments were removed: unilaterally in 3 patients (16%), bilaterally in 16 patients (84%), and laparoscopically in 89% of cases. In one patient, the horn was preserved (horno-neovaginal anastomosis). Histopathology confirmed the presence of the endometrium in uterine rudiments ipsilateral to the pain location in 75% of cases. Four patients (20%) were diagnosed with endometriosis. **Conclusions:** Recurrent pelvic pain in patients with MRKHS should prompt the diagnosis of functional uterine rudiments. The resection of symptomatic uterine horns can result in the complete resolution of pain. Patients with endometriosis are at risk of pain recurrence. In some patients strongly desirous of menstruation, horno-neovaginal anastomosis can be cautiously attempted.

## 1. Introduction

Mayer–Rokitansky–Küster–Hauser syndrome (MRKHS) is a rare congenital disorder affecting the development of the uterus and the vagina in patients with normal female phenotype, including external genitalia, and karyotype (46, XX) [1]. It combines the aplasia or hypoplasia of the uterus and vagina. Ovaries develop correctly, leading to normal ovarian function. Patients experience puberty with normal secondary sex characteristics’ development [2]. MRKHS affects 1 in 5000 female live births and is the second most common cause of primary amenorrhea [2]. There are two types of MRKHS: type 1, which involves isolated uterovaginal aplasia, while type 2 is associated with extragenital manifestations, mainly renal (renal agenesis, ectopic kidney), skeletomuscular, cardiac, vascular, ocular malformations, and hearing impairment [3]. 

The aetiology of MRKHS is unclear, but familial clustering suggests the genetic cause. Researchers using chromosomal microarray and genome-wide sequencing have successfully found significant genetic variations linked to MRKHS, including 17q12 (*LHX1*, *HNF1B*) and 16p11.2 (TBX6) deletions and sequence variations in *GREB1L* and *PAX8*, indicating a heterogeneous aetiology involving different genes [4]. In a study conducted by Brucker et al. [5], it was observed that patients diagnosed with MRKHS exhibited reduced responsiveness of endometrial stroma cells to hormones, possibly indicative of impaired hormone receptor function contributing to the syndrome’s development. Additionally, unlike healthy controls, uterine remnants in affected patients displayed distinct characteristics, including lower proliferation and a basalis-like endometrium. The findings suggest a potential role of impaired hormone receptor function in the development of MRKHS [6].

MRKHS is class U5a or U5b C4 V4 (uterine aplasia, cervical aplasia, and vaginal aplasia) according to the European Society of Human Reproduction and Embryology and the European Society for Gynaecological Endoscopy (ESHRE/ESGE) classification system of female genital tract congenital anomalies [7]. According to the American Society for Reproductive Medicine Müllerian anomalies classification (ASRM MAC) 2021 [8], MRKHS is classified as Müllerian agenesis. Both classifications include the sub-class with the presence of rudimentary horn or horns with a functional cavity (ESHRE/ESGE classification) or an atrophic uterine remnant or remnants with functional endometrium (ASRM MAC 2021). 

MRKHS is typically diagnosed in late adolescence due to primary amenorrhea. Less common presentations involve an inability to have penetrative vaginal intercourse or recurrent abdominal pain with no menstrual periods. Finally, incidental findings on child examination of external genitalia or pelvic imaging may also happen. 

While ultrasound (US) is commonly used as the first imaging technique to diagnose MRKHS due to its accessibility, magnetic resonance imaging (MRI) is considered the gold standard for the detailed imaging of Müllerian structures in complex anomalies (defined as anomalies resulting from disturbances in more than one stage of normal embryologic development and having as a result an anatomical deviation in more than one organ of the female genital tract) [9,10].

According to the literature, depending on the mode of detection (imaging, laparoscopy), between 50% and 99.8% of MRKHS patients have rudimentary uterine horns [6,11,12]. Approximately half of these patients experience recurrent abdominal or pelvic pain [13]. When surgically removed, about 50% of uterine remnants show the presence of endometrium during the histological exam [6]. 

Menstrual blood collecting in the Müllerian remnants or flowing backward into the pelvic cavity can cause hematometra or secondary endometriosis [14,15]. Alternative factors, including an upregulated epithelial-to-mesenchymal transition, might also play a role in the pathogenesis of endometriosis in MRKH syndrome [16]. Other potential causes of chronic recurrent pelvic pain in women with MRKHS may include peritoneal irritation after ovulation, ovarian cysts, myomas or adenomyosis of the Müllerian remnants, or postoperative adhesions after previous surgeries [17]. 

The most common surgical gynaecological procedure performed in patients with MRKHS is the surgical creation of a functional neovagina (neovaginoplasty); however, in the vast majority of women with this syndrome, a non-surgical treatment approach of vaginal aplasia is successful. The most challenging procedure is uterus transplantation (UTx) [4]. In patients with present unilateral or bilateral functional horns (with a cavity), the removal of uterine remnants, typically carried out through a laparoscopy and, less commonly, through a laparotomy, may be necessary. An alternative approach involves horno-vaginal anastomosis either unilaterally or after the unification of the two uterine horn remnants. 

Our study aims to report the clinical course and long-term outcomes of patients with MRKHS and symptomatic functional uterine remnants who required surgical intervention due to severe pelvic pain.

## 2. Materials and Methods

In this study, we included 20 patients with MRKHS: 8 adolescents (age < 20 years; 4 with MRKHS type 1 and 4 with MRKHS type 2) and 12 adult women (5 with MRKHS type 1 and 7 with MRKHS type 2) who underwent surgery for symptomatic uterine rudiments between July 2012 and December 2023 in the Division of Gynaecology at the Gynaecological and Obstetric Clinical Hospital, Poznan University of Medical Sciences, Poland. They were followed up until June 2024 in the hospital’s Outpatient Clinic. In all cases, pelvic US and pelvic MRI were performed before the surgery. We retrospectively analysed the patients’ records.

The statistical analysis was performed using STATISTICA10PL. In order to compare the groups, a Mann–Whitney non-parametric test was applied. *p* < 0.05 was considered statistically significant.

## 3. Results

All the patients (Table 1 and Table 2) sought medical help due to recurrent lower abdominal pain: 10 patients (50%) reported pain on both sides, 7 patients (35%) on the right side, and 3 patients (15%) on the left side. Patient no. 10, who primarily reported right-sided pain and underwent surgery to remove the ipsilateral uterine horn, after five years was readmitted with left-sided pain to remove the preserved uterine horn located on the left side. In five patients (patients no. 1, 2, 5, 6, 8), MRKHS was identified during the diagnosis of recurrent pelvic pain at the age of 12.6–14.8 (mean 13.6) years. 

Pelvic US followed by pelvic MRI revealed bilateral uterine rudiments in 17 patients (85%) and unilateral ones in 3 patients (15%) (Figure 1). The volume of the uterine horns and the volume of the endometrial cavity within the horns, evaluated by imaging and calculated with the ellipsoid formula, ranged from 0.8 to 58.3 (median 3.9) cm^3^ and from 0.03 to 12 (median 0.2) cm^3^, respectively. No significant difference in the uterine horn volume was found between symptomatic adolescents (range 0.9–23 cm^3^; median 3.8 cm^3^) and adults (range 0.8–165 cm^3^; median 6.05 cm^3^) (*p* = 0.36). We diagnosed leiomyomas of the uterine horns in two adult patients (no. 14 and 19) and left haematosalpinx and left ovarian endometriosis in patient no. 20 before surgery (Figure 2).

The age at surgery ranged from 14.7 to 42.4 (mean 25.2) years. The follow-up period ranged from 1 month to 11.6 years (the mean follow-up time was 23 months). 

In 19 patients, uterine rudiments were removed: unilaterally in 3 patients (15.8%), bilaterally in 16 patients (84.2%) (including 1 who underwent two-step removal), and laparoscopically in 89% of cases. In all but one patient, the fallopian tubes were preserved; patient no. 20 had their left fallopian tube removed due to haematosalpinx. Additionally, at the time of surgery, we identified peritoneal endometriosis (stage 2 according to the American Society for Reproductive Medicine (ASRM) revised classification of endometriosis [18]) in patient no. 15 (Figure 3), which was later confirmed histologically. In the group of 19 patients, histopathology confirmed the presence of the endometrium in uterine rudiments on the side where the pain was located in 76% of cases (22 out of 29 symptomatic horns). Five patients (26%) experienced pelvic pain despite no endometrium being histologically detected in either of the horns. The removal of uterine rudiments provided pain relief in 18 out of 19 patients (95%). In patient no. 15, further treatment with dienogest was implemented due to the pain persistence.

Patient no. 6 expressed a strong desire to menstruate and preserve at least one of the uterine remnants that were found upon imaging (Figure 4). At the age of 18.3, she underwent a diagnostic laparoscopy that revealed favourable anatomical conditions for horno-neovaginal anastomosis, with the horns located near the middle of the pelvis and easily movable. During the surgery, the patient was diagnosed with pelvic peritoneal endometriosis ASRM stage 1. Dienogest treatment was implemented. In order to prepare for the conservative surgery, the patient performed vaginal self-dilation to improve the anatomical conditions. Despite the lack of a vaginal dimple, the patient successfully achieved a 4 cm long vagina. The final surgery was performed at the age of 19.8 years. During the laparotomy, the right-sided horn was removed, while the more prominent left-sided horn was preserved and anastomosed with the neovagina after creating the proximal part of the neovaginal canal. Moreover, the progression of pelvic endometriosis from stage 1 to stage 2 was detected, and visible lesions (peritoneal and superficial ovarian) were removed. The horno-neovaginal anastomosis was successful and resulted in cyclic menstrual bleeding; however, at the follow-up visit 3 months after the surgery, the patient reported pelvic pain. After ruling out horn obstruction, dienogest treatment was reintroduced for 6 months. During the following 3 years, the patient experienced painless regular menses. Four years after the surgery, a pelvic MRI was performed due to the recurrence of pain. The results showed adenomyosis and focal ovarian endometriosis. Resuming pharmacological therapy with dienogest resulted in complete pain relief. The patient has not yet attempted to become pregnant.

Six years after the laparoscopic removal of uterine remnants, patient no. 1 complained of a painful palpable mass in one of the trocar scars but did not report any cyclic pain. A physical examination revealed a hard subcutaneous nodule in the postoperative scar medially from the right anterior superior iliac spine. An ultrasound showed a hypoechoic lesion with a diameter of 1.1 cm at the level of the aponeurosis, indicating endometriosis. Also, a right ovarian mass, 5 cm in diameter, was found. The patient was otherwise asymptomatic and did not mention any pelvic pain. She underwent a laparoscopy, during which an ovarian mature teratoma and a scar nodule were removed. Additionally, peritoneal endometriosis ASRM stage 1 was detected. A histopathological exam confirmed trocar scar endometriosis. 

## 4. Discussion

In our study, we focused solely on patients with MRKHS and symptomatic rudimentary horns. The patients were referred to our tertiary-level centre due to recurrent pelvic pain. In five patients, we identified MRKHS in the course of diagnosis of recurrent pelvic pain at the mean age of 13.6 years, which is earlier than the typical age for the diagnosis of MRKHS due to primary amenorrhea (15–16 years old). Considering the mean age of 12.2 years at menarche for the general population, the age at diagnosis was approximately 1.4 years after the onset of menses [19]. Girls experiencing cyclic or recurrent abdominal pain and a lack of menstruation during advanced puberty should be thoroughly assessed to rule out obstructive anomalies of the reproductive tract, such as imperforate hymen, transverse vaginal septum, partial vaginal aplasia, or cervical aplasia, as indicated by the American College of Obstetricians and Gynecologists [20]. Additionally, it is essential to consider Müllerian aplasia with a present rudimentary horn and functional endometrium in the differential diagnosis. 

In our study group, no significant difference in the uterine horn volume was found between symptomatic adolescents and adults. It must be emphasised that the volume of the horn cavity (and, indirectly, the horn volume) may vary, depending on the phase of the menstrual cycle and the time that has passed since the last menstrual period, due to the resorption of menstrual blood. As a result, it is challenging to establish a precise cavity volume that warrants the removal of uterine horns. Clinical symptoms, such as recurrent lower abdominal pain, are essential when determining the need for surgery to remove uterine rudiments. 

In all cases, our patients underwent either laparoscopy or laparotomy for the surgical removal of uterine horns with the preservation of fallopian tubes (except for patient no. 20, who had their left fallopian tube removed due to haematosalpinx). While laparoscopy has a low rate of complications, it is crucial to note that one of our patients experienced scar endometriosis. It is, however, a rare long-term postoperative complication and may be incidental. The implantation of endometriosis in the trocar scar in the case of patient no. 1 was likely due to the accidental transfer and implantation of disseminated tissue from when the horn with endometrium was dissected. When choosing treatment and the surgical approach, it is necessary to consider the extent of the operation. The simultaneous removal of the fallopian tubes is a topic of discussion as it may reduce ovarian vasculature and reserve. Conversely, maintaining fallopian tubes could present a risk of adnexal mass formation or future malignancy. The literature recommends removing both uterine rudiments, even if one is non-functional. However, in a study conducted by Fedele et al. [21], the second rudimentary horn, if present, was not removed to avoid compromising ovarian vascularisation. Preserving asymptomatic uterine rudiments without a functional endometrium could pose future risks for patients, as inactive horns may eventually become symptomatic. As we reported in our study, for patients who experienced pelvic pain, removal of their uterine remnants resulted in pain resolution, even though no endometrium was detected, nor any other discernible cause for their pain before surgery. Moreover, in cases with one symptomatic uterine horn with the primary intent to leave the other non-functional one, the benefits of removing both at once should be considered to avoid consecutive surgeries. A patient could be readmitted years later with contralateral pain, as in the case of the patient in our cohort. Preserving an asymptomatic horn, even if there is no sign of functional endometrium, can be a risk for the future and expose the patient to additional surgery and possible complications.

In our study, we diagnosed four patients (20%) with pelvic endometriosis. These patients seemed to have a larger volume of uterine rudiments than the patients without endometriosis, but the small cohort limits the possibility to conduct comparative statistical analyses. Further studies involving larger cohorts of patients are necessary to analyse this association. 

In order to remove endometriotic lesions, we used laparoscopic excision and ablation (electrocoagulation). An alternative method would be to treat superficial peritoneal endometriosis (SPE) with laser vaporisation [22].

MRKHS patients with endometriosis may contest the retrograde menstruation origin theory of SPE. Some of them may lack the anatomical connection between the endometrium and the peritoneal cavity due to tubal agenesis, hypoplasia, or tubal segmentary atresia. This could be validated by the case of patient no. 1 from our study: in this patient, SPE was diagnosed during the second laparoscopy, 7 years post bilateral horn removal. Nevertheless, none of our patients were diagnosed with fallopian tube agenesis. In a study published by Oppelt et al. in 2006 [1], in the group of 53 patients with MRKHS, only 1 patient had bilateral agenesis of the tubes and 6% had unilateral agenesis. In the study published by Oppelt et al. in 2012 [11], normal adnexa were found in 248/284 (87.3%) of patients with MRKHS. The presence of haematosalpinx in patient no. 20 shows that retrograde menstruation does occur in MRKHS women with functional horns.

A study published by Pietzsch M. et al. [23] involved exclusively MRKHS patients who underwent the surgical, laparoscopy-assisted creation of a neovagina. In 85.5% of these patients (265/310), the presence of uterine rudiments was confirmed during surgery. 89 out of 196 women (45%) who had rudiments removed (18.9% of the total study group) had a history of cyclic abdominal pain, but whether this was due to functional Müllerian remnants or other causes had not been clarified because the follow-up of these patients was not reported. In 80 of 196 patients (40.8%) who had rudiments removed, the presence of endometrial tissue in the rudiments was confirmed by a histopathological examination. None of the patients was diagnosed with endometriosis, though the volume of the largest uterine rudiment reached 184.3 cm^3^, which is larger than any uterine horn in our study. 

Steinmaher et al. [24] analysed endometriotic tissue and the endometrium of uterine remnants from patients with MRKHS, who underwent the surgical, laparoscopy-assisted creation of a neovagina using the modified Vecchietti technique combined with the removal of uterine remnants. The median age of the patients at the time of surgery was 23.5 years. The authors found that, out of the 319 patients with removed uterine remnants, none had hematometra due to endometrial proliferation, but 9 (including 4 patients younger than 21 years) had endometriosis (3.1%), 5 patients had adenomyosis within the uterine remnants, 2 patients had peritoneal endometriosis, and 2 had ovarian endometriosis. Three out of nine patients (33%) reported cyclic abdominal pain. No information is available about the pelvic or abdominal pain experienced by the remaining 310 patients in this study. This confirms that, in patients with MRKHS and uterine rudiments without symptoms, there is no need to perform a diagnostic laparoscopy or remove uterine remnants, as the risk of endometriosis is low. The authors assumed that the incidence of endometriosis in patients with MRKHS and a functioning endometrium might be at least as high as in the general population of women of a reproductive age, which is about 10–15%, but, in the general cohort of women with MRKHS, it is considerably lower. In our study, the incidence of endometriosis was 20%; however, this included only patients with recurrent pelvic pain and functioning uterine remnants. 

Fedele et al. [21] presented the biggest-to-date group of patients with MRKHS and bilateral uterine horns who underwent horno-neovaginal anastomoses. The study included eight adolescents (mean age at the time of surgery 16.5 years). In seven cases, the second rudimentary horn (when present) was left, and in one, it was removed due to reported dysmenorrhea. Two out of eight patients had stage 1 pelvic endometriosis. During the 2–12-year follow-up, all patients remained free of postoperative complications, continued menstruating, and engaged in sexual activity without experiencing dyspareunia. In our cohort, for patient no. 6, who underwent horno-neovaginal anastomosis, the long-term postoperative course was less favourable. Following three years of painless menstruation, the patient presented with progressive dysmenorrhea and was diagnosed with more advanced endometriosis (including adenomyosis). 

According to Fedele et al. [21], laparoscopically assisted horno-vestibular anastomosis may be considered a safe and effective therapeutic option in patients with MRKHS, similar to patients with cervicovaginal aplasia. This approach offers potential psychological benefits as the inability to menstruate can contribute to a negative self-evaluation of femininity [20]. Horno-neovaginal anastomosis provides hope to patients to maintain not only their menstruation but also their gestational potential. The question of whether this hope is true or false remains unanswered. Even in patients with congenital cervical aplasia, successful obstetric outcomes are a significant challenge [25]. To this date, no reports have been published on successful pregnancy in anastomosed uterine remnants. Our experience suggests that the risk of endometriosis development or progression and its potential influence on pregnancy outcomes should also be highlighted in the decision-making process concerning horno-neovaginal anastomosis.

Additional investigations and long-term follow-ups are important to determine the safety and effectiveness of horno-neovaginal anastomosis in adolescents with MRKHS and functional uterine rudiments. 

Our study was a retrospective analysis, which we consider to be a major limitation. It was conducted in a single tertiary-level centre with a relatively small study group. 

## 5. Conclusions

Recurrent pelvic pain in adolescents and adults with MRKHS should prompt the diagnosis of functional uterine rudiments. The resection of symptomatic uterine horns can result in the complete resolution of symptoms in patients without secondary endometriosis. In some patients strongly desirous of menstruation, preserving the functional uterine horn (horno-neovaginal anastomosis) can be cautiously attempted. Regardless of age, MRKHS patients with preserved uterine remnants should be closely monitored.

## Figures and Tables

**Figure 1 jcm-13-06767-f001:**
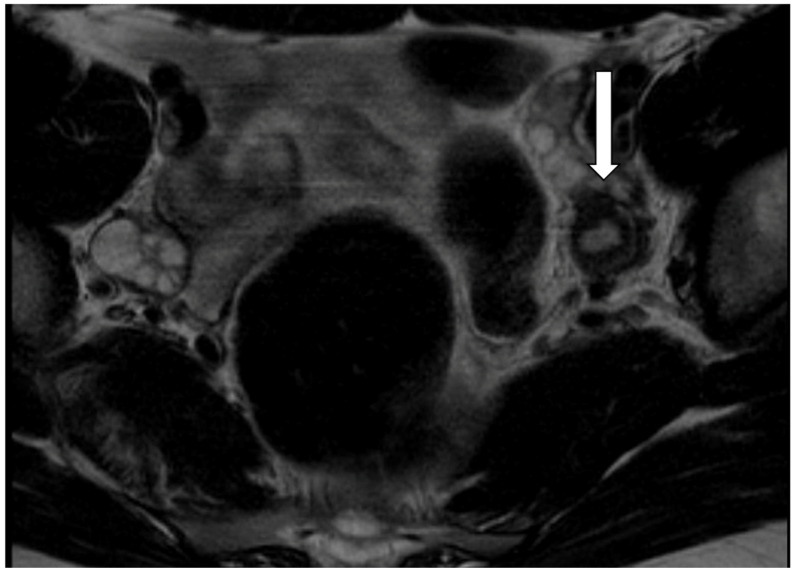
Patient 1. Magnetic resonance imaging (MRI) (T2-TSE-TRA) showing left uterine rudiment (arrow).

**Figure 2 jcm-13-06767-f002:**
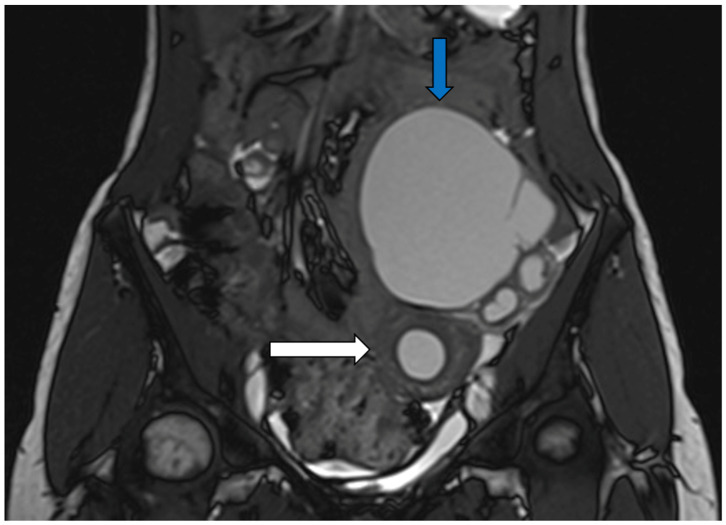
Patient 20. MRI (T2-TSE-COR) showing uterine rudiment (white arrow) and left haematosalpinx (blue arrow).

**Figure 3 jcm-13-06767-f003:**
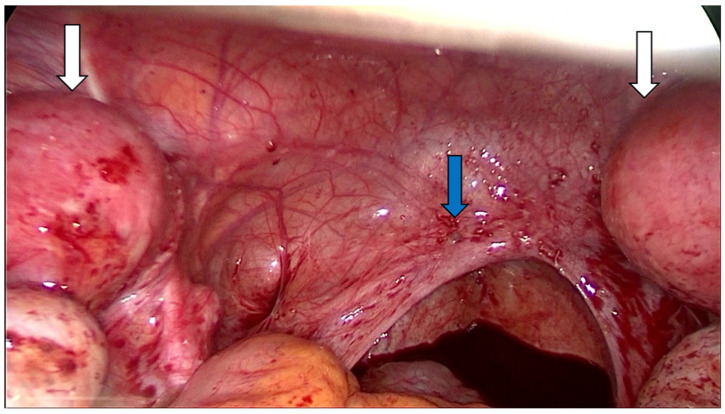
Patient 15. Bilateral uterine rudiments (white arrows) and peritoneal endometriosis (numerous clear vesicular lesions; adhesions; and the blue arrow indicates a blue lesion)—laparoscopic view.

**Figure 4 jcm-13-06767-f004:**
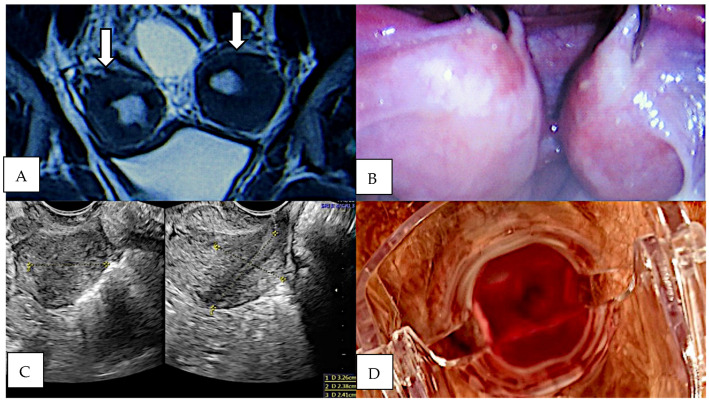
Patient 6. (**A**) MRI (T2-TSE-COR) showing bilateral uterine rudiments (arrows). (**B**) Bilateral uterine rudiments and peritoneal endometriosis—laparoscopic view. (**C**) Transvaginal pelvic ultrasound showing preserved left-sided horn, 2 years post horn removal. (**D**) Vaginoscopy.

**Table 1 jcm-13-06767-t001:** Characteristics of eight adolescent patients with MRKHS diagnosed with symptomatic uterine rudiments.

No	Age at Surgery (yrs)	Type of MRKHS and Coexisting Anomalies	Volume (cm^3^)				Pain Location Before Surgery	Type of Surgery	Presence of Endometrium in Uterine Horns Confirmed in Histopathology	Endometriosis (ASRM Stage)	Pelvic Pain After Surgery
			R Horn	R Cavity	L Horn	L Cavity					
1	15.6 22.6	1	20.6	1.4	3.7	0.2	RL	LSC: bilateral resection LSC: removal of ovarian teratoma, scar endometriosis excision	R (+) L (+)	Yes Stage 1 peritoneal	No
2	16.3 18.8	2 Bilateral conductive hearing loss	3	0	2.4	0	RL	LSC: bilateral resection Modified Wharton neovagino-plasty	R (−) L (−)	No	No
3	18.1	2 Skeletal: lumbarisation S1, Anal atresia, congenital heart disease	2.7	0	1.9	0.03	RL	LSC: bilateral resection	R (−) L (−)	No	No
4	18.3	1	13.4	2.3	0.9	0	R	LSC: bilateral resection	R (+) L (−)	No	No
5	17.3	1	7.8	0.2	4.5	0.2	RL	LSC: bilateral resection	R (+) L (+)	No	No
6	18.3 19.8	2 Gothic palate Severe malocclusion of teeth	13.6	0.9	23	0.9	RL	Diagnostic LSC LT: resection of the right horn, neovagino-plasty left horno-neovaginal anastomosis	R (+) L (+)	Yes progression from stage 1 (peritoneal) to stage 3 (peritoneal, ovarian, adenomyosis)	Yes
7	18.1	1	3.9	0.2	1.1	0	R	LSC: bilateral resection	R (+) L (−)	No	No
8	14.7	2 Scoliosis	3.5	0.2	3.9	0.2	RL	LSC: bilateral resection	R (+) L (+)	No	No

R—right; L—left; ASRM—American Society for Reproductive Medicine; LSC—laparoscopy; and LT—laparotomy.

**Table 2 jcm-13-06767-t002:** Characteristics of twelve adult patients with MRKHS diagnosed with symptomatic uterine rudiments.

No	Age at Surgery (yrs)	Type of MRKHS and Coexisting Anomalies	Volume (cm^3^)				Pain Location Before Surgery	Type of Surgery	Presence of Endometrium in Uterine Horns Confirmed in Histopathology	Endometriosis (ASRM Stage)	Pelvic Pain After Surgery
			R Horn	R Cavity	L Horn	L Cavity					
9	25.4 30.3	2 Hiatal hernia of the diaphragm	3.1	0	4	0.1	L	Modified Wharton neovagino-plasty LSC: bilateral resection	R (−) L (+)	No	No
10	26.4 31.7	1	2.5	0.03	2.9	0	R L	LSC: resection of the right horn LSC:resection of the left horn	R (+) L (+)	No	No
11	34.2	1	4.5	0.3	0.8	0	R	LSC: bilateral resection	R (+) L (−)	No	No
12	34.1	1	9.6	0.1	5.2	0	R	LSC: bilateral resection	R (+) L (−)	No	No
13	22	2 Left kidney agenesis, Klippel–Feil syndrome	3.5	0.2	0	0	R	LT: resection of the right horn	R (+)	No	No
14	42.4	2 Left kidney agenesis	165 *	0	0	0	R	LT: resection of the right horn with leiomyoma	R (−)	No	No
15	23.1	2 Right kidney agenesis	22.5	2.4	16.1	1.1	RL	LSC: bilateral resection	R (+) L (+)	Yes stage 2 (peritoneal)	Yes
16	31.9	2 Atrial septal aneurysm	7.2	0.9	2.3	0	R	LSC: bilateral resection	R (+) L (−)	No	No
17	23.9	2 Bilateral hemimelia Right pelvic kidney	0	0	6.9	0.06	L	LSC: unilateral resection	L (+)	No	No
18	24.1	1	8.3	0.8	7.7	0.8	RL	LSC: bilateral resection	R (−) L (−)	No	No
19	40.7	1	2.3	0	0	0	L	LSC: bilateral resection	R (−) L (−)	No	No
20	30.7	2 Anal atresia	12	0	58.3	12	RL	LSC: bilateral resection	R (+) L (+)	Yes stage 3 (left haemato-salpinx and left ovarian)	No

* With 8 cm leiomyoma. R—right; L—left; ASRM—American Society for Reproductive Medicine; LSC—laparoscopy; and LT—laparotomy.

## Data Availability

The original contributions presented in this study are included in this article; further inquiries can be directed to the corresponding authors.

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
