# Peer review of "Symptomatic Uterine Rudiments in Adolescents and Adults with Mayer–Rokitansky–Küster–Hauser Syndrome (MRKHS): Management and Outcomes"

_jcm, 2024, doi:10.3390/jcm13226767_

Round 1
Reviewer 1 Report
Comments and Suggestions for Authors
This is a well-structured and clearly written manuscript. The design of your study is strong, and the results are presented in a clear and logical manner. I found the methodology appropriate, and the discussion effectively ties together the findings. The flow of the manuscript is seamless, making it easy to follow. Additionally, sharing personal retrospective cohorts on the management and outcomes of this syndrome is very valuable, given its rarity. The long follow-up period reported by the authors is also a notable strength.
Line 81: consider adding a more relevant reference about endometriosis to support your sentence.
Line 144: was endometriosis confirmed histologically after surgery? Please add this information if available
Figure 4: it seems that legend referred to the D image (speculum image) is missing.
Comments:
-In patients with Rokitansky syndrome (MRKH) who have rudimentary uterine horns containing functional endometrial tissue, retrograde menstruation is unlikely due to the absence of a complete anatomical connection between the endometrium and the peritoneal cavity. In many cases, the Fallopian tubes may be present but underdeveloped, dysfunctional, or not in continuity with the lumen of the rudimentary horns, making direct passage of menstrual blood into the peritoneum difficult. Therefore, the explanation of superficial peritoneal endometriosis (SPE) origin and development must be probably seeked somewhere else. Solely SPE may cause chronic pelvic pain, therefore it is essential to remove all the macroscopic lesions when visible.
-To minimize the risk of accidental transfer and implantation of endometriosis fragments in the trocar wound, it would be advisable to treat superficial peritoneal endometriosis with laser surgery. This approach would completely avoid contact with the tissue and reduce the chances of dissemination. If possible, could you specify the technique used to remove the endometriotic lesions? This information could help readers understand how these recurrences developed. In this article https://doi.org/10.3390/jcm13061722 all these aspects are clearly explained, consider citing to further strenghten the manuscript.
Reviewer 2 Report
Comments and Suggestions for Authors
This manuscript aimed to study the clinical course of symptomatic uterine rudiments in adolescents and adults with Mayer-Rokitansky-Küster-Hauser syndrome (MRKHS).
Well presented data and well written manuscript.
Here are my comments to the authors:
Abstract:
The introduction of the abstract is very short—at least one sentence about the topic of the manuscript is to be added.
In the discussion section: The authors wrote (In five patients, we identified MRKHS in the course of diagnosis of recurrent pelvic pain at the mean age of 13.6 years which is earlier than the typical age for the diagnosis of MRKHS due to primary amenorrhea (15-16 years old).).
Could you please explain this? What about girls' earlier puberty now compared to before? Could you please explain and provide evidence?
Line 250, Write 3 as three.
Tables 1 and 2: Please correct locatior to location in the table (pain location before surgery).
How would you explain the absence of endometrium in uterine horns confirmed in histopathology?
Figure 4: Please revise the legend of the figure and correct it. D is missing.
Author Response
Point-by-point response to Comments and Suggestions for Authors
Comment 1: The introduction of the abstract is very short—at least one sentence about the topic of the manuscript is to be added.
Revisions: updated abstract
Background: Women with aplastic uterus (ESHRE/ESGE classification) or Müllerian agenesis (ASRM MAC 2021 might present with functional uterine remnants. Our study aimed to report the clinical course of symptomatic uterine rudiments in adolescents and adults with Mayer-Rokitansky-Küster-Hauser syndrome (MRKHS).
Comment 2: How would you explain the absence of endometrium in uterine horns confirmed in histopathology?
Response 2: During the surgery the horns were morcellated or dissected with monopolar electrode to facilitate removal through the laparoscopic port. The coagulation might result in destruction of the residual amount of endometrium, especially in case of presence of single epithelia endometrial cells and affect the histopathologic evaluation.
Comment 3: In the discussion section: The authors wrote (In five patients, we identified MRKHS in the course of diagnosis of recurrent pelvic pain at the mean age of 13.6 years which is earlier than the typical age for the diagnosis of MRKHS due to primary amenorrhea (15-16 years old).).Could you please explain this? What about girls' earlier puberty now compared to before? Could you please explain and provide evidence?
Response 3: Considering the mean age at menarche 12.2 years for the general population of girls, this age at diagnosis was approximately 1.4 years after the onset of menses.
Revisions:
Lines 201-203
Biro FM, Pajak A, Wolff MS, Pinney SM, Windham GC, Galvez MP, Greenspan LC, Kushi LH, Teitelbaum SL. Age of Menarche in a Longitudinal US Cohort. J Pediatr Adolesc Gynecol. 2018 Aug;31(4):339-345.
- Additional comments
Tables 1 and 2: Please correct locatior to location in the table (pain location before surgery). updated in the manuscript
Line 250: write 3 as three
updated in the manuscript
Figure 4: Please revise the legend of the figure and correct it. D is missing.
updated text in the manuscript: D. Vaginoscopy.
